# Identifying the Target Population for Primary Respiratory Syncytial Virus Two-Step Prevention in Infants: Normative Outcome of Hospitalisation Assessment for Newborns (NOHAN)

**DOI:** 10.3390/vaccines10050729

**Published:** 2022-05-06

**Authors:** Marine Jourdain, Mehdi Benchaib, Dominique Ploin, Yves Gillet, Etienne Javouhey, Come Horvat, Mona Massoud, Marine Butin, Olivier Claris, Bruno Lina, Jean-Sebastien Casalegno

**Affiliations:** 1Laboratoire de Virologie, Institut des Agents Infectieux, Laboratoire Associé au Centre National de Référence des Virus des Infections Respiratoires, Hospices Civils de Lyon, 69004 Lyon, France; mjourdain@lhopitalnordouest.fr (M.J.); bruno.lina@chu-lyon.fr (B.L.); 2Service de Médecine et de la Reproduction, Hôpital Femme Mère Enfant, Hospices Civils de Lyon, 69500 Bron, France; mehdi.benchaib@chu-lyon.fr; 3Service de Réanimation Pédiatrique et d’Accueil des Urgences, Hôpital Femme Mère Enfant, Hospices Civils de Lyon, 69500 Bron, France; dploin@me.com (D.P.); yves.gillet@chu-lyon.fr (Y.G.); etienne.javouhey@chu-lyon.fr (E.J.); come.horvat@chu-lyon.fr (C.H.); 4CIRI, Centre International de Recherche en Infectiologie, Team VirPatH, Université Lyon, Inserm, U1111, Université Claude Bernard Lyon 1, CNRS, UMR5308, École Normale Supérieure de Lyon, 69007 Lyon, France; 5Hospices Civils de Lyon, Service de Gynécologie-Obstétrique, Hôpital Femme-Mère-Enfant, 69000 Bron, France; mona.massoud@chu-lyon.fr; 6Service de Néonatologie et de Réanimation, Hôpital Femme-Mère-Enfant, Hospices Civils de Lyon, Néonatale, 69500 Bron, France; marine.butin@chu-lyon.fr (M.B.); olivier.claris@chu-lyon.fr (O.C.)

**Keywords:** RSV, bronchiolitis, predictive score, lower respiratory tract infection, primary prevention, monoclonal antibody, public health, vaccines

## Abstract

Background: Respiratory syncytial virus (RSV) is the leading cause of acute respiratory infection- related hospitalisations in infants (RSVh). Most of these infants are younger than 6 months old with no known risk factors. An efficient RSVh prevention program should address both mothers and infants, relying on Non-Pharmaceutical (NPI) and Pharmaceutical Interventions (PI). This study aimed at identifying the target population for these two interventions. Methods: Laboratory-confirmed RSV-infected infants hospitalised during the first 6 months of life were enrolled from the Hospices Civils de Lyon birth cohort (2014 to 2018). Clinical variables related to pregnancy and birth (sex, month of birth, birth weight, gestational age, parity) were used for descriptive epidemiology, multivariate logistic regression, and predictive score development. Results: Overall, 616 cases of RSVh in 45,648 infants were identified. Being born before the epidemic season, prematurity, and multiparity were independent predictors of RSVh. Infants born in January or June to August with prematurity and multiparity, and those born in September or December with only one other risk factor (prematurity or multiparity) were identified as moderate-risk, identifying the mothers as candidates for a first-level NPI prevention program. Infants born in September or December with prematurity and multiparity, and those born in October or November were identified as high-risk, identifying the mothers and infants as candidates for a second-level (NPI and PI) intervention. Conclusions: It is possible to determine predictors of RSVh at birth, allowing early enrollment of the target population in a two-level RSV prevention intervention.

## 1. Introduction

Respiratory syncytial virus (RSV) infection is the predominant cause of lower respiratory tract infection (LRTI) in infants [1,2]. In developed countries, RSV represents the leading cause of hospitalisation during the first year of life [3], with up to 3% of all infants hospitalised every year [4]. Severe cases are more frequent in infants younger than 3 months [5] and in infants with known risk factors such as prematurity, lung or heart diseases, and immunodeficiency [6]. However, most infants hospitalised with RSV LRTI have no known risk factors [2,7,8].

Although RSV is now recognised as a major health burden worldwide, preventive approaches remain limited. Non-Pharmaceutical Interventions (NPI) such as hand hygiene, breastfeeding, and avoiding exposure to smoke or persons with acute respiratory illness are effective in reducing the risk of infection with respiratory viruses in infants [9,10]. The implementation of NPI on a large scale during the COVID-19 pandemic, which strongly reduced the RSV epidemic [11], advocates for a stronger implementation of NPI. Pharmaceutical Intervention (PI) is currently limited to the only licensed drug, Palivizumab, a humanised monoclonal antibody that shows some benefit in preventing RSV disease in high-risk infants [12]. More recently, encouraging progress was made using a fusion protein nanoparticle vaccine administered to pregnant women [13] and long half-life monoclonal antibodies administered to newborns [14]. Considering the numerous PIs for RSV prevention currently being evaluated in clinical trials, there is a reasonable hope that in coming years, PI will be broadly recommended to the general population [15].

Nevertheless, successful implementation of the future prevention programs largely relies on their capacity to target the mothers of infants with the higher risk. Recently, it has been reported that prophylaxis regimens, adjusted for regional variations in terms of RSV seasonality, may improve protection compared to the implementation of nationally recommended regimens [15,16]. 

Predictive models can help adjust the preventive action to the target population [17]. Regarding the risk of RSV-associated hospitalizsation (RSVh), most of the published predictive models have focused on high-risk preterm infants [18,19,20,21,22]. Only Houben et al. proposed a simple prediction rule that can identify infants at risk of RSV LRTI, but the cohort was limited in size and the score designed only for clinical use [23].

The Normative Outcome of Hospitalisation Assessment for Newborns (NOHAN) strategy proposed herein aimed to adjust multi-level mother-infant interventions according to the risk of RSVh at the general population level. The cut-off values can be adapted to balance the resources available and the effectiveness of the preventive approach. 

## 2. Materials and Methods

### 2.1. Data Sources

Demographic and laboratory data were collected retrospectively. First, the administrative registry of all infants born in the University Hospitals of Lyon (Hospices Civils de Lyon, HCL) was used. Stillborn children or those living outside the region were excluded. This database includes the following variables of interest: gender (male/female), month of birth, gestational age (WG), maternal parity (primiparity/multiparity), plurality (single/multiple gestation), childbirth type (vaginal birth/caesarean section), birth weight, and geographical living area (postcode). Patients with at least one missing piece of data for any studied variable were excluded. The virology laboratory database was then used to identify, among all infants born in the HCL and hospitalised during the RSV season with acute respiratory infection symptoms, those with a respiratory sample positive for RSV [24]. 

Cases were defined as a new admission, during the first 6 months of life, to one of the conventional paediatric hospital departments of the HCL with an RSV positive sample during hospital stay. This 6-month follow-up period is in line with the preventive approaches proposed herein (NPI, maternal vaccination program, monoclonal antibodies administered at birth) which are mostly protective over the first 6 months of life [17]. Presumed nosocomial cases (i.e., cases observed during the birth stay) were not excluded considering that only a few cases were expected and that transmission from the community (mother, siblings, family) could not be ruled out.

### 2.2. Variable and Categorical Construction

A term delivery was defined as a baby born >37 weeks of gestation (WG) measured in weeks of amenorrhoea (WA). Moderate and very preterm deliveries were defined as a baby born ≥32 WG and ≤37 WG and a baby born ˂32 WG, respectively. 

Birth weight was categorised as either low or high when the value was outside two standard deviations of the weight cohort distribution per WG. Month of birth was categorised into four groups with increased incidence of RSVh during the first 6 months of life. 

The choice to aggregate these two variables was made to match the categories used in guidelines and clinical practice and to further facilitate the implementation of [25]. 

### 2.3. Cohort Construction

Overall, 45,648 infants were included in the study (2014 to 2018) from a catchment area of 1,370,678 inhabitants using public hospital registry data from the HCL (Figure 1) [26]. To ensure the stability of the score, the population studied was divided into three cohorts. The validating cohort was defined as the 2018 birth cohort (*n* = 8709 infants born between January 2018 and December 2018). The remaining cohort (2014 to 2017) was then randomly divided, with equal-sized month of birth proportion, into a training cohort (70% of the remaining cohort, *n* = 25,858) and a testing cohort (30% of the remaining cohort, *n* = 11,081). The validating cohort was exposed to the 2018/2019 RSV season, which was similar to the previous epidemics [27] (Table 1).

### 2.4. Identification of Predictors of Interest and Predictive Score Generation

To identify potential independent maternal RSVh predictors, all variables of the training cohort were entered into a multivariate backwards stepwise logistic regression model to remove predictors that did not significantly improve the fit of the logistic regression model (based on a *p*-value of 0.10). The odds ratio (OR) values obtained from the final model were rounded to the nearest integer and used to determine the score. The model was adjusted with the year of birth and the hospital of birth (Table 2).

### 2.5. Estimation of the Model’s Performance and Variance

To evaluate the optimism-corrected performance values, the real-life model’s performance was quantified using the area under the receiver operating characteristic curve (AUROC) (with 95% confidence interval) and the Brier score in the testing and validating cohorts. A k-fold (with k = 4) cross-validation approach was used to estimate the model’s variance on the training cohort. The distribution of month of birth was preserved in the split training sets obtained using the k-fold cross validation. The validating cohort corresponds to a new set of infants exposed to a new RSV epidemic and was used as an external control. 

In the absence of any similar predictive model but based on references from research on other medical predictive models, and given the potential cost-effectiveness of the preventive program, an AUROCC ≥ 0.70 was considered as evidence for good discrimination [28,29]. A test was considered significant when *p* value was lower than 0.05.

### 2.6. Determining the Cut-Off Values of the Two-Level Preventive Program

The optimal cut-off value for the first level of intervention (NPI alone) was defined a priori with minimal sensitivity and specificity values of 80% and 20%, respectively. This cut-off assumed a lower cost and effectiveness for NPI compared to PI. However, an NPI preventive program may not be maintained year-round if the perceived risk is too low. This assumption was informed by results showing that the application of NPI measures by French parents decreased over time during the COVID-19 crisis (CoviPrev survey) [30]. 

The optimal cut-off value for the second level of intervention (NPI and PI) was defined a priori with a minimal specificity value of 80%. This cut-off assumed a higher cost and effectiveness for PI that would be then more suitable for a timely seasonal administration than a year-round administration [31]. 

### 2.7. Statistical Analyses

Statistical analyses were performed using R software version 4.1.0 with the following packages (ggplot2; reshape2; doBy; caret; scoring, pROC). The caret library was used for data splitting. For the comparison of cohorts, *p* values were derived using the chi-squared test for qualitative variables and the Student’s *t*-test or ANOVA tests for quantitative variables.

### 2.8. Ethics

Parents of infants for whom hospitalisation data were used were informed of the study (aim of the study, use of anonymised data, right to refuse participation) by postal mail. After this first contact, data were anonymised. Authorisation from Scientific and Ethical Committee of Hospices Civils de Lyon (Comité Scientifique et Éthique des Hospices Civils de LYON CSE-HCL—IRB 00013204; Pr Cyrille Confavreux) was obtained on 28 September 2021.

### 2.9. Role of the Funding Source

There was no funding source for this study. The corresponding author had full access to all the data in the study and had the final responsibility for the decision to submit for publication.

## 3. Results

### 3.1. Characteristics of the Training, Testing, and Validating Cohorts

Several patients were missing data for gestational age and were excluded (572/46,220). The population studied comprised 45,648 infants: 50.7% (23,140) were boys, 16.1% (7354) were born preterm (≤37 WG), 5.8% (2641) from multiple births, 20.8% (9497) were born by caesarean section, and 43.7% (19,933) from multiparous mothers. The median (Interquartile Range [IQR]) birth weight was 3207 [2920–3580] g. Among them, 616 were hospitalised in their first 6 months of life with a laboratory-confirmed RSV infection, 53.7% (331) were boys, 22.8% (140) were born preterm (≤37 WG), 7.3% (45) from multiple births, and 73.4% (452) from multiparous mothers (Table 1). Among all cases, 0.8% (5) were suspected to be nosocomial. The characteristics of the testing and training cohorts did not significantly differ. The validating cohort did not significantly differ in terms of mean WG and birth weight when compared to the testing and training cohorts. Significant differences in the frequencies of month of birth, parity, gestation type, and childbirth type were observed between the validating cohort and the testing and training cohorts (Table 1).

### 3.2. Month of Birth Categorisation in the Population Studied

Month of birth was categorised into four groups with increased incidence of RSVh during the first 6 months of life. In the population studied 95.1% (586/616) of the cases were detected from November to February with a peak during December (Figure 2). Mean (±Standard Deviation SD) age at the time of hospitalisation was 2.1 (±1.4) months.

Incidence was 13.5 infants hospitalised in the first 6 months of life per 1000 infants (95% Confidence Interval (CI) [12.0; 15.0]). The incidence of RSVh in the first 6 months of life was higher for those born just before the seasonal epidemic, from October to November (41.3 cases per 1000 infants (95% CI [37.0; 46.0]) compared to those born in September and December (23.0 cases per 1000 infants (95% CI [20.0; 27.0]), as well as to those born in January and June to August (7.53 cases per 1000 infants (95% CI [6.0; 9.0]), and those born February to May (0.81 cases per 1000 infants (95% CI [0.0; 1.0]; Figure 2).

In the model analysis, the month of birth was aggregated into these four groups of respective increased RSVh incidence. The choice to aggregate this variable was done to facilitate the implementation of the score in the patient and healthcare community in the context of pregnancy.

### 3.3. Maternal Predictors of Interest Associated with Increased Risk of RSVh (Training Cohort)

The multivariate logistic regression analysis in the training cohort retrieved three variables significantly associated with hospitalisation: month of birth, gestational age, and parity. Considering the month of birth, being born during October and November was associated with the highest risk of RSVh (OR 46.35, 95% CI [24.40; 102.73], *p* < 0.001). Considering the gestational age, being born ˂32 WG was associated with a higher risk of RSVh (OR 3.26, 95% CI [1.80; 5.52], *p* < 0.001). Considering parity, multiparity was associated with a higher risk of RSVh (OR 3.89, 95% CI [3.07; 4.97], *p* < 0.001; Table 2). These variables, along with the year of birth and the hospital of birth, were retained in the final predictive model and used for the score construction.

### 3.4. Performance of the RSVh Predictive Score in the Testing and Validating Cohorts 

The final score value ranged from 0 to 53. The mean (±SD) score was 16.98 (±16.42) in the testing cohort and 18.41 (±19.64) in the validating cohort (external control). 

The k-fold validation approach used in the training cohort to estimate the model’s variance found a mean AUROC value of 0.791 (95% CI [0.788; 0.793]) for the four split training cohorts compared with a mean AUROC value of 0.821 (95% CI: [0.791; 0.850]) for the validating cohort. The overlap between the two confidence intervals indicates a low variance.

The Hosmer–Lemehow chi-squared test used to assess the goodness of fit (*p* value < 0.001) and the Brier score measured (value = 0.0135) in the validating cohort both indicated a good performance of the predictive model.

### 3.5. Identifying the Target Population for the Two-Level Maternal-Infant Preventive Program

The optimal cut-off values of the RSVh predictive score for the first and second levels of intervention were 14 and 28, respectively. The maternal population with an RSVh predictive score below the first cut-off value (14) of intervention was defined as a low-risk group (Table 3). It encompasses all the mothers giving birth during the months of February to May and the deliveries occurring during the months of January or June to August with no more than one risk factor (prematurity or multiparity). 

The maternal population with an RSVh predictive score above the first cut-off value (>14) and below the second cut-off value (<28) of intervention was defined as a moderate-risk group. It encompasses the deliveries occurring during the months of January or June to August with prematurity and multiparity and the deliveries in September or December with only one other risk factor (prematurity or multiparity).

The maternal population with an RSVh predictive score above the second cut-off value (>28) was defined as a high-risk group. It encompasses the deliveries in September or December with prematurity and multiparity and all deliveries in October or November (Table 3).

When applied to the validating cohort, the predictive score identified 53% of infants as low-risk (incidence 1.72 cases/1000 infants), 32% of infants as moderate-risk (incidence 18.92/1000 infants), and 15% of infants as high-risk (incidence 46.28/1000 infants; Table 3).

## 4. Discussion

In the present study, an original but simple approach was used to identify the target population for a two-level preventive mother-infant intervention program based on maternal and newborn risk factors.

The clinical predictors identified herein were already described in previous reports [19,20,21]. Month of birth is by far the strongest predictor of RSVh in the general birth cohort. This finding is consistent with numerous previous observations reporting that infants born before the RSV peak month have the highest RSVh admission rates [32,33]. Prematurity under 37 WG and multiparity were also previously associated with increased RSVh risk [34,35].

The present approach differs from other RSVh prediction models [19,20,21,22]. The novelty herein is the use of RSVh risk factors that can be identified early in the pregnancy (month of birth and multiparity) or at the infant’s birth (WG). It allows the inclusion of target mothers and then infants in a prevention program tailored to the infant’s risk of RSVh. The first level entails a reinforced behaviour change program (BCP) to promote good hygiene practices (hand hygiene, breastfeeding, avoiding exposure to smoke and persons with acute respiratory illness) and education on bronchiolitis. The second level is a pharmaceutical prevention program that will integrate future licensed preventive drug interventions. These BCP and PI can be introduced early during pregnancy follow-up so that the different healthcare workers involved in the pregnancy, childbirth, and postnatal periods can repeatedly promote them to the target population.

A major strength of this study was that all the diagnoses were PCR-confirmed with only a minor number of missing values. Indeed, identifying cases based on diagnosis codes, such as admissions for bronchiolitis, is neither fully specific nor sensitive. On one hand, bronchiolitis can be caused by other respiratory viruses such as rhinoviruses, metapneumoviruses, and influenza viruses [36]. On the other hand, RSV admissions could be related to disease codes other than bronchiolitis (sepsis, acute otitis). According to the hospital’s local protocol, all infants below the age of 1 year old hospitalised with acute respiratory infection are tested for RSV during the RSV epidemic season [24]. A sensitivity analysis performed over one RSV season (2016/2017) showed that 83.4% of all hospitalisations for bronchiolitis were due to RSV (PCR-confirmed), while 11.5% were related to other viruses (5% did not have any sample tested) [37]. The second major strength of this work is that a complete case analysis was performed, as there was only a small proportion of infants excluded for missing data. In addition, dividing the population studied into three cohorts allowed a robust estimation of the score’s performance.

There are, however, some limitations to this work. First, the cost-effectiveness of the future licensed preventive treatments remains to be determined. One consequence of the COVID-19 crisis may be a better integration and promotion of NPI in the prevention of viral respiratory diseases. Although more costly, the future drugs in development will likely represent a more efficient preventive approach if dedicated to high-risk infants. Therefore, this two-step strategy is likely to be considered in the near future. Another limitation is that some known risk factors (exposure to smoke, parent’s education level, mother’s age) [35,38] were not present in the database and therefore not tested in the model. Palivuzimab use was also not measured in this study. According to the French guidelines, Palivuzimab is recommended for premature babies born before 29 WG or born before 32 WG with risk factors. This did not represent more than 2% of the present birth cohort. Given that extremely premature infants are a small subgroup of the general population, and that extreme prematurity was still a higher risk factor than preterm and term births, it was assumed that the general model was not significantly affected by the use of Palivizumab.

## 5. Conclusions

Using a hospital birth cohort, the NOHAN strategy allowed to determine strong maternal and newborn predictors of RSVh risk. By using this strategy, future parents could be enrolled early during pregnancy follow-up in a health-related BCP. The pregnant women could then be proposed a vaccine boost, or neutralizing monoclonal antibodies could be administered to the newborns. As demonstrated herein, the thresholds for triggering each level of intervention can be adjusted to the local epidemiology, the resources available, and the evolving evidence concerning the cost-efficiency of the future interventions. Stakeholders, healthcare professionals, and policy makers should acknowledge this opportunity when designing the future of RSV prevention programs.

## Figures and Tables

**Figure 1 vaccines-10-00729-f001:**
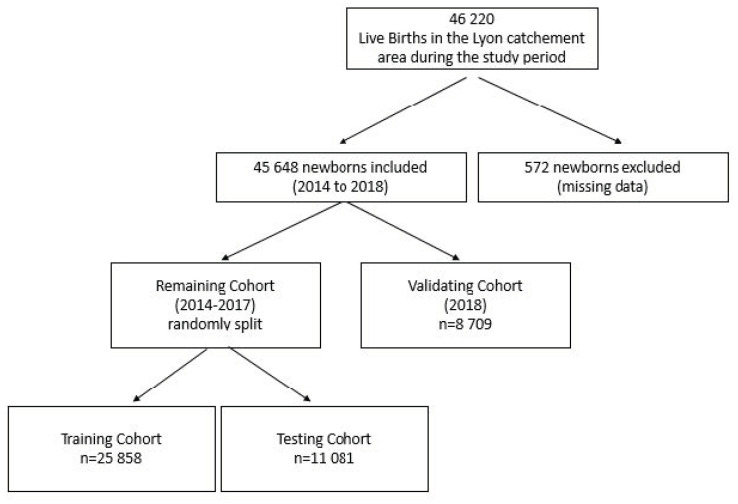
Flowchart of the study.

**Figure 2 vaccines-10-00729-f002:**
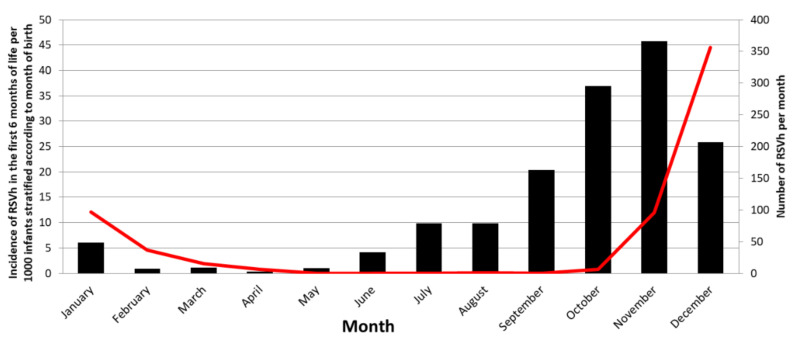
Incidence of infants hospitalised during the first six months of life according to month of birth and number of hospitalisations per month in the studied population. Red line: Number of RSVh per month; Black column: Incidence of RSVh in the first 6 months of life per 1000 infants stratified according to month of birth.

**Table 1 vaccines-10-00729-t001:** Demographic and clinical characteristics of the training, testing, and validating cohorts.

	Training Cohort 2014–2017 *n* = 25,858	Testing Cohort 2014–2017 *n* = 11,081	*p* value	Validating Cohort 2018 *n* = 8709	*p* Value
Gender, n (%) Male	13,159 (50.9)	5631 (50.8)	0.90	4350 (50.0)	0.30
Month of birth, n (%)					
January	2196 (8.5)	959 (8.7)	0.36	771 (8.9)	<0.0001
February	1935 (7.5)	841 (7.6) 4 47 (7.9%)		664 (7.6)	
March	2152 (8.3)	865 (7.8)		708 (8.1)	
April	2036 (7.8)	900 (8.1)		753 (8.7) 3	
May	2268 (8.8)	913 (8.2)		753 (8.7)	
June	2155 (8.3)	934 (8.4)		814 (9.4)	
July	2209 (8.5)	995 (9.0)		821 (9.4) 3902	
August	2201 (8.5)	922 (8.3)		822 (9.5)	
September	2120 (8.2)	962 (8.7)		793 (9.1)	
October	2252 (8.7)	933 (8.4)		633 (7.3)	
November	2208 (8.5)	920 (8.3)		608 (7.0)	
December	2126 (8.2)	937 (8.5)		569 (6.5)	
Weeks of gestation, n (%) Median (IQR**)	39 [38–40]	39 [38–40]	0.54	39 [38–40]	0.06
>37 weeks	21,649 (83.7) 1405 (2.55	9273 (83.7)	0.77	7372 (84.6)	0.16
≥32 and ≤37 weeks	3633 (14.0)	1573 (14.2)		1173 (13.5)	
˂32 weeks	576 (2.2)	235 (2.1)		164 (1.9)	
Parity, n (%)					
Primiparity	14,759 (57.1)	6258 (56.5)	0.28	4698 (53.9)	<0.0001
Multiparity	11,099 (42.9)	4823 (43.5)		4011 (46.1)	
Gestation, n (%)					
Single gestation	24,322 (94.1)	10,427 (94.1)	0.89	8258 (94.8)	0.03
Multiple gestation	1536 (6.0)	654 (5.9)		451 (5.2)	
Childbirth type, n (%)					
Vaginal birth	20,224 (78.2)	8694 (78.5)	0.60	7233 (83.0)	<0.0001
Caesarean section	5634 (21.8)	2387 (21.5)		1476 (17.0)	
Birth weight (grams), n (%)					
Median (IQR)	3260 [2920–3585]	3270 [2920–3590]	0.54	3260 [2920–3580]	0.78
Low Z score	464 (1.8)	199 (1.8)	0.70	229 (2.6)	<0.0001
Normal Z score	24,631 (95.2)	10,573 (95.4)		8336 (95.7)	
Macrosomic Z score	763 (3.0)	309 (2.8)		144 (1.7)	
Year of birth, n (%)					
2014	6386 (24.7)	2754 (24.9)	0.03	0 (0)	NA
2015	6554 (25.3)	2648 (24.9)		0 (0)	
2016	6377 (24.7)	2807 (25.3)		0 (0)	
2017	6541 (25.3)	2872 (25.9)		0 (0)	
2018	0 (0)	0 (0)		8709 (100)	

Abbreviations: IQR, interquartile range, NA: not appropriate for testing.

**Table 2 vaccines-10-00729-t002:** Multivariate analysis of the risk factors for RSV-associated hospitalisation during the first 6 months of life and predictive variables.

Variable	Multivariable Analysis			Final Model			Score
	OR	[95% CI]	*p* Value	OR	[95% CI]	*p* Value	OR Round up
Sex							
Female	Reference						
Male	1.08	[0.85; 1.30]	0.66				
Month of birth							
February to May	Reference				Reference		**0**
January, June to August	9.25	[4.75; 20.84]	<0.00010	9.23	[5.10; 23.97]	<0.0001	**9**
September, December	22.82	[11.78; 51.21]	<0.000100	22.76	[11.75; 51.07]	<0.0001	**23**
October, November	46.35	[24.40; 102.73]	<0.0001	46.37	[24.41; 102.78]	<0.0001	**46**
Gestational age Median (IQR)							
>37 weeks	Reference				Reference		**0**
≥32 and ≤37 weeks	1.64	[1.21; 2.20]	0.00	1.71	[1.29; 2.24]	0.00	**2**
<32 weeks	3.26	[1.80; 5.52]	<0.0001	3.43	[1.93; 5.68]	<0.0001	**3**
Parity							
Primiparity	Reference				Reference		**0**
Multiparity	3.89	[3.07; 4.97]	<0.0001	3.88	[3.06; 4.95]	<0.0001	**4**
Gestation							
Simple gestation	Reference						
Multiple gestation	1.16	[0.77; 1.70]	0.45				
Birth weight (%)							
Normal Z score	Reference						
Low Z score	1.37	[0.53; 2.87]	0.46				
Macrosomic Z score	1.42	[0.81; 2.30]	0.19				

Abbreviations: OR, odds ratio; CI, confidence interval; NS, non-significant.

**Table 3 vaccines-10-00729-t003:** Characteristics of the low-, moderate-, and high-risk groups according to the predictive model.

	Low Risk Group	Moderate Risk Group	High Risk Group
Risk Factors	Any delivery February to Mayanddeliveries in January or June to August with only one other risk factor (prematurity or multiparity)	Deliveries in January or June to August with prematurity and multiparity, anddeliveries in September or December with only one other risk factor (prematurity or multiparity)	Any delivery in October to November and deliveries in September or December with two risk factors (prematurity and multiparity)
Preventive Intervention	Standard	Non-Pharmaceutical Intervention Education Program	Non-Pharmaceutical Intervention Education Program And Pharmaceutical Intervention Maternal Vaccine (term), or Monoclonal antibody (term, preterm)
Population proportion *	53% (4643/8709)	32% (2748/8709)	15% (1318/8709)
Incidence RSVh /1000 **	1.72/1000	18.92/1000	46.28/1000

* Calculated on the validating cohort. ** Incidence of infants hospitalised in the first 6 months of life per 1000 infants.

## Data Availability

Aggregated data are available on reasonable request.

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
