# Peer review of "Identifying the Target Population for Primary Respiratory Syncytial Virus Two-Step Prevention in Infants: Normative Outcome of Hospitalisation Assessment for Newborns (NOHAN)"

_vaccines, 2022, doi:10.3390/vaccines10050729_

Round 1

Reviewer 1 Report

Respiratory syncytial virus (RSV) is one of the leading causes of acute respiratory infection-related hospitalizations in infants, the elderly, and immunocompromised people. The authors have performed hospital-recorded pediatric birth month and RSV RT-PCR confirmed diagnosis data-based epidemiological investigation to identify the risk factors of infant RSV disease hospitalization. The author's approach is novel involving the use of RSV hospitalization risk factors that can be identified early in the pregnancy based on the month of birth and multiparity or gestational age in weeks at the infant’s birth. Although the number of infant hospitalized were only 616 of 45648, the analysis clearly supports the risk factors considered in implementing the prediction model. 

The next step for the authors would be to develop a funding proposal for the nationwide implementation of a prevention program for mothers and at-risk infants for RSV hospitalization prevention programs.

Edits required

Line 313 : external fundin - please correct spelling to "funding"

Author Response

Response to Reviewers

We would like to thank the reviewers for their thoughtful comments and efforts towards improving our manuscript. We have accepted all suggestions on spelling and wording from both reviewers and have modified the manuscript on this way.

For reviewer 2 : we do not have the exact percentage of bronchiolitis due to RSV not PCR-confirmed, for this study, but as said line 285, on precedent study made on RSV season 2016/2017 : only 5% did not have any sample tested.We can supposed this percentage is  substantially the same.

Reviewer 2 Report

Jourdain et al. have developed an efficient method for reducing RSV hospitalizations by characterizing both mothers and infants to identify the target population, for eventual interventions. Laboratory-confirmed RSV-infected infants hospitalized during the first 6 months of life were enrolled during 4 years in a birth cohort and clinical variables related to pregnancy and birth were collected. 616 cases of RSV hospitalized infants out of 45,648 infants were identified. Infants born in January or June to August with prematurity and multiparity, and those born in September or December with only one other risk factor (prematurity or multiparity) were identified as moderate risk, identifying the mothers as candidates for a first level prevention program. Infants born in September or December with prematurity and multiparity, and those born in October or November were identified as high-risk, identifying the mothers and infants as candidates for a second level intervention.

This study is a thoughtful, straight-forward study of the relationship of birth date and the two major risk factors, prematurity and multiparity, as predictors of hospitalization due to RSV infection. The authors divided the cohort into three groups for training, testing, and confirming their results. They present their results convincingly, with one exception. Table 3 is alluded to in the text, but there is no Table 3.

A few suggestions on spelling and wording are marked on the attached manuscript.

Author Response

(The authors gave the same response as above.)
